# Truly privacy-preserving federated analytics for precision medicine with multiparty homomorphic encryption

David Froelicher [1], Juan R. Troncoso-Pastoriza [1], Jean Louis Raisaro [2,3], Michel A. Cuendet[4], Joao Sa Sousa[1], Hyunghoon Cho[5], Bonnie Berger[5,6,7], Jacques Fellay [2,8] & Jean-Pierre Hubaux [1✉]

Using real-world evidence in biomedical research, an indispensable complement to clinical trials, requires access to large quantities of patient data that are typically held separately by multiple healthcare institutions. We propose FAMHE, a novel federated analytics system that, based on multiparty homomorphic encryption (MHE), enables privacy-preserving analyses of distributed datasets by yielding highly accurate results without revealing any intermediate data. We demonstrate the applicability of FAMHE to essential biomedical analysis tasks, including Kaplan-Meier survival analysis in oncology and genome-wide association studies in medical genetics. Using our system, we accurately and efficiently reproduce two published centralized studies in a federated setting, enabling biomedical insights that are not possible from individual institutions alone. Our work represents a necessary key step towards overcoming the privacy hurdle in enabling multi-centric scientific collaborations.

[1] Laboratory for Data Security, EPFL, Lausanne, Switzerland. [2] Precision Medicine Unit, Lausanne University Hospital, Lausanne, Switzerland. [3] Data Science Group, Lausanne University Hospital, Lausanne, Switzerland. [4] Precision Oncology Center, Lausanne University Hospital, Lausanne, Switzerland. [5] Broad Institute of MIT and Harvard, Cambridge, MA, USA. [6] Computer Science and AI Laboratory, MIT, Cambridge, MA, USA. [7] Department of Mathematics, MIT, Cambridge, MA, USA. [8] School of Life Sciences, EPFL, Lausanne, Switzerland. ✉email: jean-pierre.hubaux@epfl.ch

A key requirement for fully realizing the potential of precision medicine is to make large amounts of medical data interoperable and widely accessible to researchers. Today, however, medical data are scattered across many institutions, which renders centralized access and aggregation of such data challenging, if not impossible. The challenges are not due to the technical hurdles of transporting high volumes of heterogeneous data across organizations but to the legal and regulatory barriers that make the transfer of patient-level data outside a healthcare provider complex and time-consuming. Moreover, stringent data protection and privacy regulations (e.g., General Data-Protection Regulation (GDPR)[1]) strongly restrict the transfer of personal data, including even pseudonymized data, across jurisdictions.

Federated analytics (FA) is emerging as a new paradigm that seeks to address the data governance and privacy issues related to medical-data sharing[2–4]. FA enables different healthcare providers to collaboratively perform statistical analyses and to develop machine-learning models, without exchanging the underlying datasets. Only aggregated results or model updates are transferred. In this way, each healthcare provider can define its own data governance and maintain control over the access to its patient-level data. FA offers opportunities for exploiting large and diverse volumes of data distributed across multiple institutions. These opportunities can facilitate the development and validation of artificial intelligence algorithms that yield more accurate, unbiased, and generalizable clinical recommendations, as well as accelerate novel discoveries. Such advances are particularly important in the context of rare diseases or medical conditions, where the number of affected patients in a single institution is often not sufficient to identify meaningful statistical patterns with enough statistical power.

The adoption of FA in the medical sector, despite its potential, has been slower than expected. This is in large part due to the unresolved privacy issues of FA, related to the sharing of model updates or partial data aggregates in cleartext. Indeed, despite patient-level data not being transferred between the institutions engaging in FA, it has been shown that the model updates (or partial aggregates) themselves can, under certain circumstances, leak sensitive personal information about the underlying individuals, thus leading to re-identification, membership inference, and feature reconstruction[5,6]. Our work focuses on overcoming this key limitation of existing FA approaches. We note that limited data interoperability across different healthcare providers is another potential challenge in deploying FA; this, in practice, can be surmounted by harmonizing the data across institutions before performing the analysis.

Several open-source software platforms have recently been developed to provide users streamlined access to FA algorithms[3,7,8]. For example, DataSHIELD[7] is a distributed data analysis and a machine-learning (ML) platform based on the open-source software R. However, none of these platforms address the aforementioned problem of indirect privacy leakages that stem from their use of "vanilla" federated learning. Hence, it remains unclear whether these existing solutions are able to substantially simplify regulatory compliance, compared to more conventional workflows that centralize the data[9–11], if the partial aggregates and model updates could still be considered as personal identifying data[5,6,12–14].

More sophisticated solutions for FA, which aim to provide end-to-end privacy protection, including for the shared intermediate data, have been proposed[15–25]. These solutions use techniques such as differential privacy (diffP)[26], secure multiparty computation (SMC), and homomorphic encryption (HE). However, these techniques often achieve stronger privacy protection at the expense of accuracy or computational efficiency, thus limiting their applicability. Existing diffP techniques for FA, which

prevent privacy leakage from the intermediate data by adding noise to it before sharing, often require prohibitive amounts of noise, which leads to inaccurate models. Furthermore, there is a lack of consensus around how to set the privacy parameters for diffP in order to provide acceptable mitigation of inference risks in practice[27]. SMC and HE are cryptographic frameworks for securely performing computation over private datasets (pooled from multiple parties in the context of FA, in an encrypted form) without any intermediate leakage, but both come with notable drawbacks. SMC incurs a high network-communication overhead and has difficulty scaling to a large number of data providers (DPs). HE imposes high storage and computational overheads and introduces a single point of failure in the standard centralized setup, where a single party receives all encrypted datasets to securely perform the joint computation. Distributed solutions based on HE[21–23,28] have also been proposed to decentralize both the computational burden and the trust, but existing solutions address only simple calculations (e.g., counts and basic sample statistics) and are not suited for complex tasks.

Here, we present FAMHE, an approach, based on multiparty homomorphic encryption (MHE)[29], to privacy-preserving FA, and we demonstrate its ability to enable efficient federated execution of two fundamental workflows in biomedical research: Kaplan–Meier survival analysis and genome-wide association studies (GWAS). MHE is a recently proposed multiparty computation framework based on HE; it combines the power of HE to perform computation on encrypted data without communication between the parties, with the benefits of interactive protocols, which can simplify certain expensive HE operations. Building upon the MHE framework, we introduce an approach to FA, where each participating institution performs local computation and encrypts the intermediate results by using MHE; the results are then combined (e.g., aggregated) and distributed back to each institution for further computation. This process is repeated until the desired analysis is completed. Contrary to diffP-based approaches that rely on obfuscation techniques to mitigate the leakage in intermediate results, by sharing only encrypted intermediate results, FAMHE provides end-to-end privacy protection, without sacrificing accuracy. By sharing only encrypted information, our approach guarantees that, whenever needed, a minimum level of obfuscation can be applied only to the final result in order to protect it from inference attacks, instead of being applied to all intermediate results. Furthermore, FAMHE improves over both SMC and HE approaches by minimizing communication, by scaling to large numbers of DPs, and by circumventing expensive noninteractive operations (e.g., bootstrapping in HE). Our work also introduces a range of optimization techniques for FAMHE, including optimization of the local vs. collective computation balance, ciphertext packing strategies, and polynomial approximation of complex operations; these techniques are instrumental in our efficient design of FAMHE solutions for survival analysis and GWAS.

We demonstrate the performance of FAMHE by replicating two published multicentric studies that originally relied on data centralization. These include a study of metastatic cancer patients and their tumor mutational burden (TMB)[30], and a host genetic study of human immunodeficiency virus type 1 (HIV-1)-infected patients[31]. By distributing each dataset across multiple DPs and by performing federated analyses using our approach, we successfully recapitulated the results of both original studies. Our solutions are efficient in terms of both execution time and communication, e.g., completing a GWAS over 20K patients and four million variants in <5 h. In contrast to most prior work on biomedical FA, which relied on artificial datasets[15,17,23,32], our results closely reflect the potential of our approach in real application settings. Furthermore, our approach has the potential

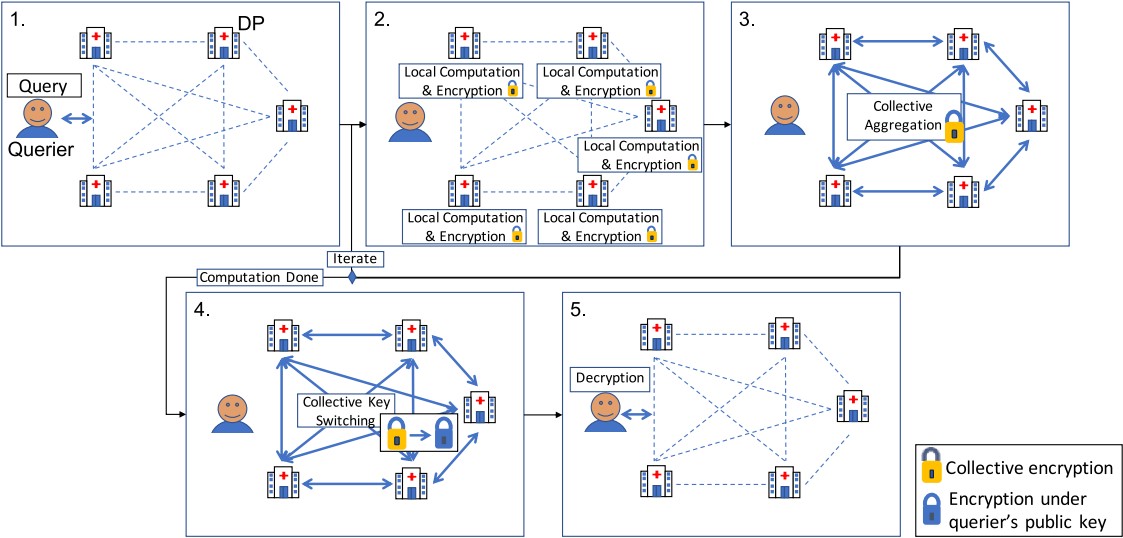

**Fig. 1 System Model and FAMHE workflow.** All entities are interconnected (dashed lines) and communication links at each step are shown by thick arrows. All entities (data providers (DPs) and querier) are honest but curious and do not trust each other. In 1. the querier sends the query (in clear) to all the DPs who (2.) locally compute on their cleartext data and encrypt their results with the collective public key. In 3. the DPs' encrypted local results are aggregated. For iterative tasks, this process is repeated (Iterate). In 4. the final result is then collectively switched by the DPs from the collective public key to the public key of the querier. In 5. the querier decrypts the final result.

to simplify the requirements for contractual agreements and the obligations of data controllers that often hinder multicentric medical studies, because data processed by using MHE can be considered anonymous data under the GDPR[12]. Our work shows that FAMHE is a practical framework for privacy-preserving FA for biomedical workflows and it has the power to enable a range of analyses beyond those demonstrated in this work.

## Results

**Overview of FAMHE.** In FAMHE, we rely on MHE to perform privacy-preserving FA by pooling the advantages of both interactive protocols and HE and by minimizing their disadvantages. In particular, by relying on MHE and on the distributed protocols for FA proposed by Froelicher et al.[24], our approach enables several sites to compute on their local patient-level data and then encrypt (Local Computation & Encryption in Fig. 1) and homomorphically combine their local results under MHE (Collective Aggregation (CA) in Fig. 1). These local and global steps can be repeated (Iterate in Fig. 1), depending on the analytic task. At each new iteration, participating sites use the encrypted combination of the results of the previous iteration to compute on their local data without the need for decryption, e.g., gradient-descent steps in the training of a regression model. The collectively encrypted and aggregated final result is eventually switched (Collective Key Switching in Fig. 1) from encryption under the collective public key to encryption under the querier's public key (the blue lock in Fig. 1) such that only the querier can decrypt. The use of MHE ensures that the secret key of the underlying HE scheme never exists in full. Instead, the control over the decryption process is distributed across all participating sites, each one holding a fragment of the decryption key. This means that all participating sites have to agree to enable the decryption of any piece of data and that no single entity alone can decrypt the data. As described in System and Threat Model in the "Methods" section, FAMHE is secure in a passive adversarial model in which all but one DPs can be dishonest and collude among themselves.

FAMHE builds upon optimization techniques for enabling the efficient execution of complex iterative workflows: (1) by relying

on edge computing and optimizing the use of computations on the DPs' cleartext data; (2) by relying on the packing ability of the MHE scheme to encrypt a vector of values in a single ciphertext such that any computation on a ciphertext is performed simultaneously on all the vector values, i.e., Single Instruction, Multiple Data (SIMD); (3) by further building on this packing property to optimize the sequence of operations by formatting a computation output correctly for the next operation; (4) by approximating complex computations such as matrix inversion (i.e., division) by polynomial functions (additions and multiplications) to efficiently compute them under HE; and (5) by replacing expensive cryptographic operations by lightweight interactive protocols. Note that FAMHE avoids the use of centralized complex cryptographic operations that would require a more conservative parameterization and would result in higher computational and communication overheads (e.g., due to the use of larger ciphertexts). Therefore, FAMHE efficiently minimizes the computation and communication costs for a high-security level. We provide more details of our techniques in the "Methods" section.

We implemented FAMHE based on Lattigo[33], an open-source Go library for multiparty lattice-based homomorphic encryption cryptography. We chose the security parameters to always ensure high 128-bit-level security. We refer to the "Methods" section for a detailed configuration of FAMHE used in our experiments.

To demonstrate the performance of FAMHE, we developed efficient FA solutions based on FAMHE and our optimization techniques for two essential biomedical tasks: Kaplan–Meier survival analysis and GWAS. We present the results of these solutions on real datasets from two peer-reviewed studies that were originally conducted by centralizing the data from multiple institutions.

**Multicentric Kaplan–Meier survival analysis using FAMHE.** Kaplan–Meier survival analysis is a widely used method to assess patient's response (i.e., survival) over time to a specific treatment. For example, in a recent study, Samstein et al.[30] demonstrated that the TMB is a predictor of clinical responses to immune checkpoint inhibitor (ICI) treatments in patients with metastatic

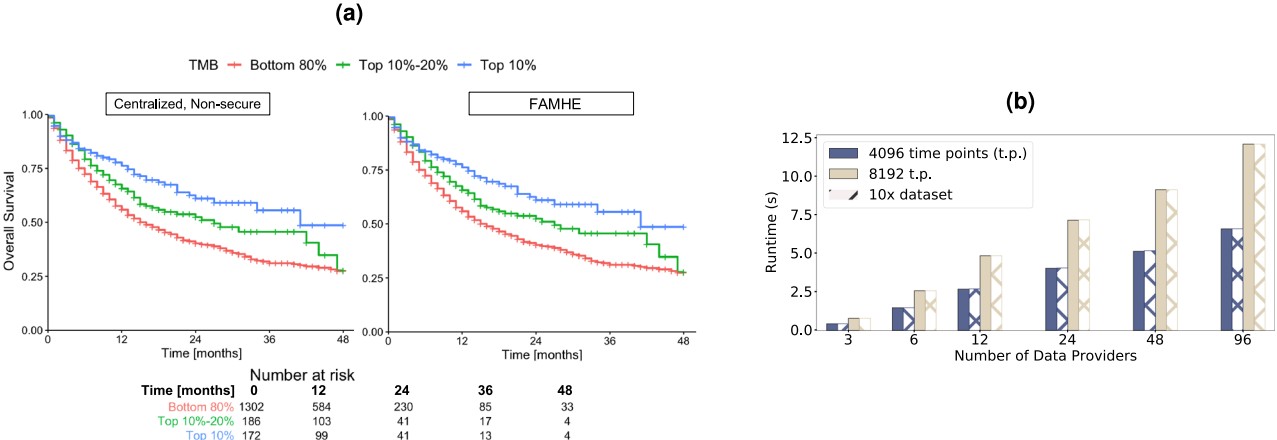

**Fig. 2 Secure and distributed reproduction of a survival-curve study. a** Survival curves generated in a centralized nonsecure manner and with FAMHE on the data used by Samstein et al.[30]. TMB stands for tumor mutational burden. With FAMHE, the original data are split among three data providers, and the querier obtains exact results. The table in **a** displays the number of patients at risk at a specific time. The exact same numbers are obtained with the centralized, nonsecure solution and with FAMHE. **b** FAMHE execution time for the computation of one (or multiple) survival curve(s) with a maximum of 8192 time points. For both the aggregation and key switching (from the collective public key to the querier's key), most of the execution time is spent in communication (up to 98%), as the operations on the encrypted data are lightweight and parallelized on multiple levels, i.e., among the data providers and among the encrypted values.

cancers. To obtain this conclusion, they computed Kaplan–Meier overall survival (OS) curves of 1662 advanced-cancer patients treated with ICI, and that are stratified by TMB values. OS was measured from the date of first ICI treatment to the time of death or the last follow-up. In Fig. 2, we show the survival curves obtained from the original centralized study (Centralized, Non-secure) and those obtained through our privacy-preserving federated workflow of FAMHE executed among three DPs. Note that for FAMHE, to illustrate the workflow of federated collaboration, we distributed the dataset across the DPs, each hosted on a different machine. FAMHE's analysis is then performed with each DP having access only to the locally held patient-level data, thus closely reflecting a real collaboration setting that involves independent healthcare centers. As a result, our federated solutions circumvent the privacy risks associated with data centralization in the original study. We observed that FAMHE produces survival curves identical to those of the original nonsecure approach. By using either approach, we are able to derive the key conclusion that the benefits of ICI increase with TMB.

In Fig. 2b, we show that FAMHE produces exact results while maintaining computational efficiency, as the computation of the survival curves shown in Fig. 2a is executed in < 12 s, even when the data are scattered among 96 DPs. We also observe that the execution time is almost independent of the DPs' dataset size, as the same experiment performed on a 10× larger dataset (replicated 10×) takes almost exactly the same amount of time. We show that FAMHE's execution time remains below 12 s for up to 8192 time points. We note that, in this particular study, the number of time points (instants at which an event can occur) is smaller than 200, due to the rounding off of survival times to months. In summary, the FAMHE-based Kaplan–Meier estimator produces precise results and scales efficiently with the number of time points, each DPs' dataset size, and the number of DPs. We remark that the hazard ratio, which is often computed in survival-curve studies, can be directly estimated by the querier, based on the final result[34]. It is also possible to compute the hazard ratios directly by following the general workflow of FAMHE described in Fig. 1. This requires the training of proportional-hazard regression models that are closely related to generalized linear models[35] that our GWAS solution also utilizes.

**Multicentric GWAS using FAMHE**. GWAS are a fundamental analysis tool in medical genetics that identifies genetic variants that are statistically associated with given traits, such as disease status. GWAS have led to numerous discoveries about human health and biology, and efforts to collect larger and more diverse cohorts to improve the power of GWAS. Their relevance to diverse human populations continues to grow. As we progress toward precision medicine and genetic sequencing becomes more broadly incorporated into routine patient care, large-scale GWAS that span multiple medical institutions will become increasingly more valuable. Here, we demonstrate the potential of FAMHE to enable multicentric GWAS that fully protects the privacy of patients' data throughout the analysis.

We evaluated our approach on a GWAS dataset from McLaren et al.[31]; they studied the host genetic determinants of HIV-1 viral load in an infected population of European individuals. It is known that the viral load observed in an asymptomatic patient after primary infection positively correlates with the rate of disease progression; this is the basis for the study of how host genetics modulates this phenotype. We obtained the available data for a subset of the cohort including 1857 individuals from the Swiss HIV Cohort Study, with 4,057,178 genotyped variants. The dataset also included 12 covariates that represent ancestry components, which we also used in our experiments to correct for confounding effects. To test our federated-analysis approach, we distributed, in a manner analogous to the survival analysis experiments, the GWAS dataset across varying numbers of DPs.

Following the approach of McLaren et al.[31], we performed GWAS using linear regression of the HIV-1 viral load on each of the more than four million variants, always including the covariates. To enable this large-scale analysis in a secure and federated manner, we developed two complementary approaches based on our system: FAMHE-GWAS and FAMHE-FastGWAS. FAMHE-GWAS performs exact linear regression and incurs no loss of accuracy, whereas FAMHE-FastGWAS achieves faster runtime through iterative optimization at a small expense of accuracy. We believe that both modes are practical and that the choice between them would depend on the study setting. Importantly, both solutions do not reveal intermediate results at any point during the computation, and any data exchanged between the DPs to facilitate the computation are always kept

hidden by collective encryption. We also emphasize that the DPs in both solutions utilize their local cleartext data and securely aggregate encrypted intermediate results, following the workflow presented in Fig. 1.

Both our solutions use a range of optimized computational routines that we developed in this work to carry out the sophisticated operations required in GWAS by using MHE. In FAMHE-GWAS, we exploit the fact that the same set of covariates are included in all regression models by computing once the inverse covariance matrix of the covariates, then for each variant computing an efficient update to the inverse matrix to reflect the contribution of each given variant. Our solution employs efficient MHE routines for each of these steps, including matrix inversion. In FAMHE-FastGWAS, we first subtract the covariate contributions from the phenotype by training once a linear model including only the covariates. We then train in parallel univariate models for all four million variants. We perform this step efficiently by using the stochastic gradient-descent algorithm implemented with MHE. Taken together, these techniques illustrate the computational flexibility of FAMHE and its potential to enable a wide range of analyses. Further details of our solutions are provided in the "Methods" section.

We compare FAMHE-GWAS and FAMHE-FastGWAS against (i) Original, the centralized nonsecure approach adopted by the original study, albeit on the Swiss HIV Cohort Study dataset, (ii) Meta-analysis[36], a solution in which each DP locally and independently performs GWAS to obtain summary statistics that are then shared and combined (through the weighted $Z$ test) across DPs to produce a single statistic for each variant that represents its overall association with the target phenotype, and (iii) Independent, a solution in which a DP uses only its part of the dataset to perform GWAS. For all baseline approaches, we used the PLINK[36] software to perform the analysis (see "Methods" section for the detailed procedure). Note that Meta-analysis can also be securely executed by first encrypting each DP's local summary statistics then following the FA workflow presented in Fig. 1.

The Manhattan plots visualizing the GWAS results obtained by each method are shown in Fig. 3a. Both our FAMHE-based methods produced highly accurate outputs that are nearly indistinguishable from the Original results. Consequently, our methods successfully implicated the same genomic regions with genome-wide significance found by Original, represented by the strongest associated single-nucleotide polymorphisms (SNPs) rs7637813 on chromosome 3 (nominal $P = 7.2 \times 10^{-8}$) and rs112243036 on chromosome 6 ($P = 7.0 \times 10^{-21}$). Notably, both these SNPs are in close vicinity to the two strongest signals reported by the original study[31]: rs1015164 at a distance of 9 kbp and rs59440261 at a distance of 42 kbp, respectively. The former is found in the major histocompatibility complex region, and the latter is near the *CCR5* gene; both have established connections to HIV-1 disease progression[31]. Although the two previous SNPs were not available in our data subset to be analyzed, we reasonably posit that our findings capture the same association signals as in the original study, related through linkage disequilibrium. Regardless, we emphasize that our federated-analysis results closely replicated the centralized analysis of the same dataset we used in our analysis.

In contrast, the Meta-analysis approach, although successfully applied in many studies, severely underperformed in our experiments by reporting numerous associations that are likely spurious. We believe this observation highlights the limitation of meta-analyses when the sample sizes of individual datasets are limited. Similarly, the Independent approach obtained noisy results, which was further compounded by the issue of limited statistical power (for results obtained by every DP, see

Supplementary Fig. 4). We complement these comparisons with Table 1 that quantifies the error in the reported negative logarithm of $P$ value ($-\log_{10}(P)$), as well as the regression weights ($\mathbf{w}$), for all of the considered approaches compared to Original. We observed that FAMHE-FastGWAS yields an average absolute error always smaller than $10^{-2}$, which ensures accurate identification of association signals. FAMHE-GWAS further reduces the error by roughly a factor of three to obtain even more accurate results. Whereas Meta-analysis and Independent approaches result in considerably larger errors.

FAMHE scales efficiently in all dimensions: number of DPs, samples, and variants (Fig. 4). As displayed by Fig. 4a, FAMHE's runtime decreases when the workload is distributed among more DPs, and it is below 1 h for a GWAS jointly performed by 12 DPs on more than 4 million variants with FAMHE-FastGWAS. It also shows that in a wide-area network, where the bandwidth is halved (from 1 Gbps to 500 Mbps) and the delay doubled (from 20 to 40 ms), FAMHE execution time increases by a maximum of 26% over all experiments. FAMHE's execution time grows linearly with the number of patients (or samples) and variants (Fig. 4c, d). In all experiments, the communication accounts for between 4 and 55% of FAMHE's total execution time. As described in the "Methods" section, FAMHE computes the $P$ values of multiple (between 512 and 8192) variants in parallel, due to the SIMD property of the crypto scheme and is further parallelized among the DPs and by multithreading at each DP. FAMHE is therefore highly parallelizable, i.e., doubling the number of available threads would almost halve the execution time. Finally, FAMHE-GWAS, which performs exact linear regression, further reduces the error (by a factor of 3× compared to FAMHE-FastGWAS), but its execution times are generally higher than FAMHE-FastGWAS.

These results demonstrate the ability of FAMHE to enable the execution of FA workflows on data held by large numbers of DPs who keep their data locally while allowing full privacy with no loss of accuracy. To our knowledge, no other existing approaches achieve all of these properties: the FA approaches that share intermediate analysis results in cleartext among the DPs offer limited privacy protection or when used together with diffP techniques to mitigate leakage, they sacrifice accuracy. Meta-analysis approaches yield imprecise results compared to joint analysis, especially in settings where each DP has access to small cohorts, as we have shown. According to our estimates, centralized HE-based solutions have execution times that are 1–3 orders of magnitude greater than FAMHE due to the overhead of centralized computation, as well as compute-intensive cryptographic operations required by centralized HE (e.g., bootstrapping). Finally, SMC approaches, although an alternative for a small network of 2-4 DPs, have difficulty supporting a large number of DPs, due to their high communication overhead. Note that communication of SMC scales with the combined size of all datasets, whereas FAMHE shares only aggregate-level data, thus vastly reducing the communication burden. We provide a more detailed discussion of existing solutions and estimates of their computational costs in Supplementary Note 5.

## Discussion
Here, we have demonstrated that efficient privacy-preserving federated-analysis workflows for complex biomedical tasks are attainable. Our efficient solutions for survival analysis and GWAS, based on our paradigm FAMHE, accurately reproduced published peer-reviewed studies while keeping the dataset distributed across multiple sites and ensuring that the shared intermediate data do not leak any private information. Alternative approaches based on meta-analysis or

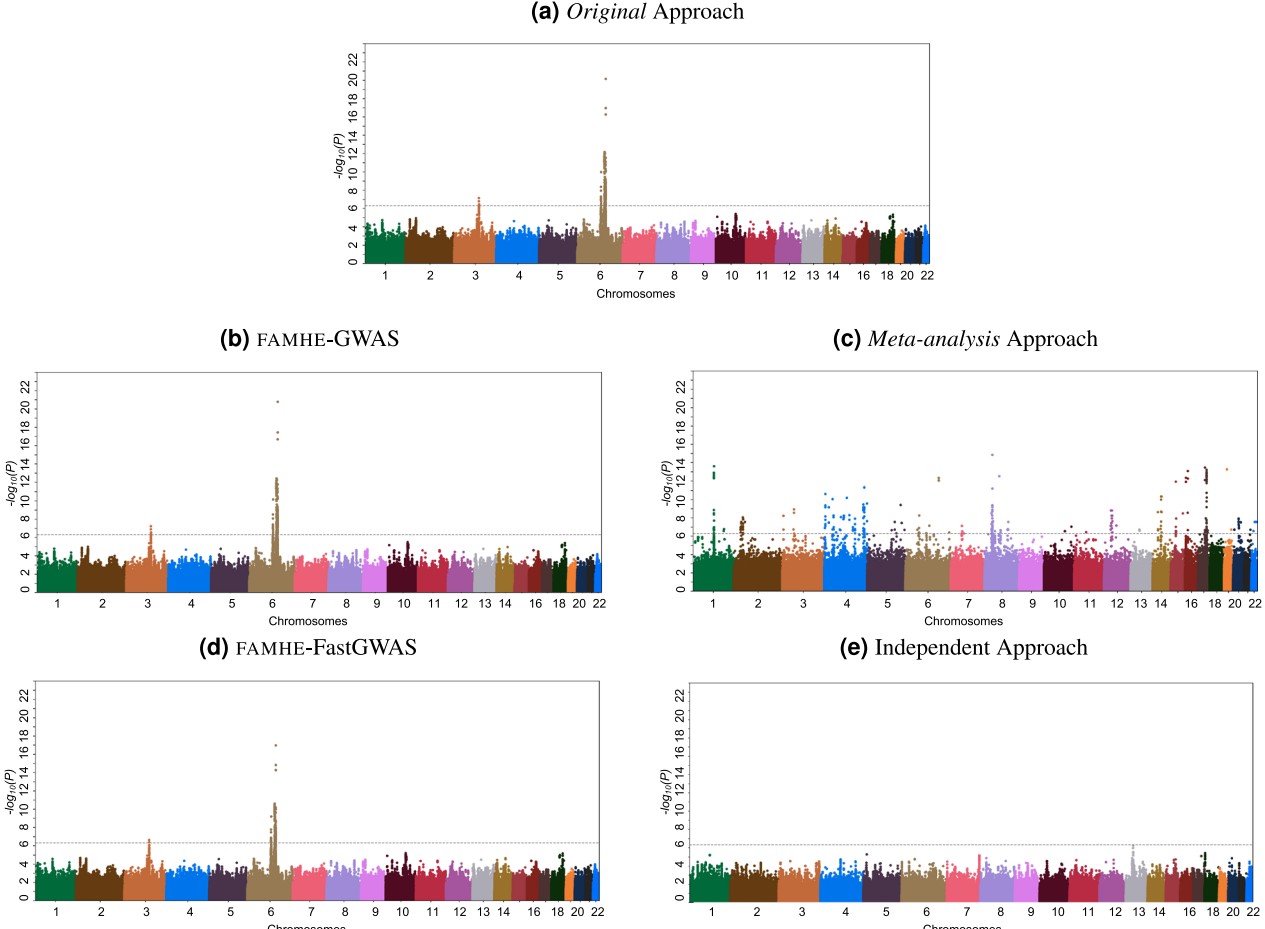

**(a)** *Original* Approach

**(b)** FAMHE-GWAS

**(c)** *Meta-analysis* Approach

**(d)** FAMHE-FastGWAS

**(e)** Independent Approach

**Fig. 3 Comparison of the GWAS results obtained with different approaches with 12 DPs (when applicable). a** Original is considered as the ground truth and is obtained on a centralized cleartext dataset by relying on the PLINK[36] software. Panels (**c**) and (**e**) are also obtained with PLINK (see "Methods" section and Supplementary Fig. 4). Panels (**b**) and (**d**) are the results obtained with FAMHE-GWAS and FAMHE-FastGWAS, respectively. In the original study and in our secure approach, genome-wide signals of association ($\log_{10}(P) < 5 \times 10^{-7}$, dotted line) were observed on chromosomes 6 and 3. The P values shown are nominal values without multiple testing correction and are obtained using standard two-sided t tests for testing whether the linear regression coefficient associated with a variant is nonzero.

**Table 1 Absolute averaged error on the logarithm of the P values ($-\log_{10}(P)$) and on the model weights (w) between Original, Independent and federated approaches.**

|  | Independent | | Meta-analysis | | FAMHE-FastGWAS | | FAMHE-GWAS | |
|---|---|---|---|---|---|---|---|---|
|  | $-\log_{10}(P)$ | w | $-\log_{10}(P)$ | w | $-\log_{10}(P)$ | w | $-\log_{10}(P)$ | w |
| **3 DPs** | | | | | | | | |
| All | 0.369 | 0.04 | 0.448 | 0.04 | $6.7e^{-3}$ | $1.5e^{-3}$ | $2.72e^{-3}$ | $7.3e^{-4}$ |
| Peaks | 4.14 | 0.055 | 7.9 | 0.19 | 0.71 | $6.61e^{-3}$ | 0.1392 | $1.88e^{-7}$ |
| **6 DPs** | | | | | | | | |
| All | 0.409 | 0.0665 | 0.45 | 0.041 | $8.3e^{-3}$ | $1.61e^{-3}$ | $2.78e^{-3}$ | $7.4e^{-4}$ |
| Peaks | 4.86 | 0.12 | 7.95 | 0.195 | 0.82 | $6.63e^{-3}$ | 0.1393 | $2.3e^{-7}$ |
| **12 DPs** | | | | | | | | |
| All | 0.425 | 0.104 | 0.453 | 0.048 | $9e^{-3}$ | $1.63e^{-3}$ | $2.79e^{-3}$ | $7.7e^{-4}$ |
| Peaks | 6.619 | 0.126 | 7.99 | 0.197 | 0.848 | $6.69e^{-3}$ | 0.1399 | $3.6e^{-7}$ |

For each number of data providers, we report the error averaged over all positions and the errors on the peaks identified with Original (see Fig. 3a). The P values shown are nominal values without multiple testing correction and are obtained using standard two-sided t tests for testing whether the linear regression coefficient associated with a variant is nonzero.

independent analysis of each dataset led to noisy results in our experiments, illustrating the benefits of our federated solutions. The fact that FAMHE led to practical federated algorithms for both the statistical calculations required by Kaplan–Meier curves and the large-scale regression tasks of GWAS reflects the ability of FAMHE to enable a wide range of other analyses in biomedical research, such as cohort exploration and the training and evaluation of disease risk prediction models.

Conceptually, FAMHE represents a novel approach to FA; it has not been previously explored for complex biomedical tasks.

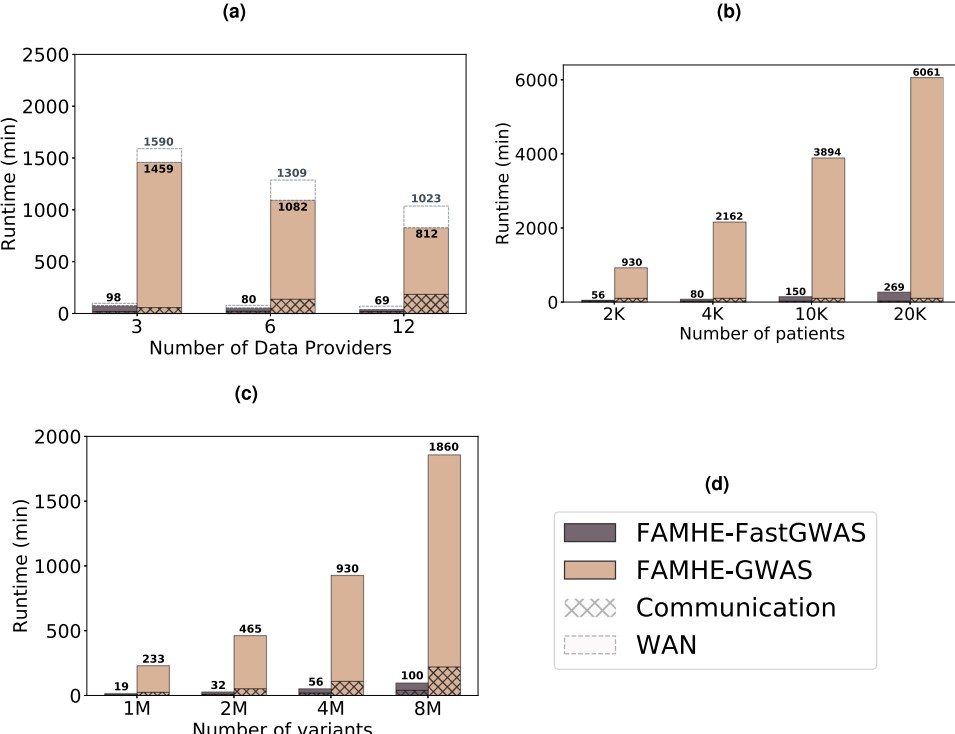

**Fig. 4 FAMHE scaling. a** FAMHE's scaling with the number of data providers, **b** with the size of the dataset, and **c** with the number of variants considered in the GWAS. Panel (**d**) is the legend box for (**a**–**c**). In (**a**), we also observe the effect of a reduced available bandwidth (from 1 Gbps to 500 Mbps) and increased communication delay (from 20 to 40 ms) on FAMHE's execution time. The original dataset containing 1857 samples and four million variants is evenly split among the data providers. By default, the number of DPs is fixed to 6.

FAMHE combines the strengths of both conventional federated-learning approaches and cryptographic frameworks for secure computation. Like federated learning, FAMHE scales to large numbers of DPs and enables noninteractive local computation over each institution's dataset (available locally in cleartext), which approach minimizes the computational and communication burdens that cryptographic solutions[17,18,21–23,37] typically suffer from. However, FAMHE draws from the cryptographic framework of MHE to enable secure aggregation and local computation of intermediate results in an encrypted form. This approach departs from the existing federated-learning solutions[2,3,7,15,16,20] that largely rely on data obfuscation to mitigate leakage in the intermediate data shared among the institutions. Our approach thus provides more rigorous privacy protection. In other words, in FAMHE, accuracy is traded off only with performance, similarly to nonsecure federated approaches, but differently from obfuscation-based solutions, FAMHE's security is absolute. We summarize our comparison of FAMHE with existing works in Supplementary Table 1, Supplementary Notes 2 and 5, and we refer to the "Methods" section for more details.

The fact that FAMHE shares only encrypted data among the DPs have important implications for its suitability to regulatory compliance and its potential to catalyze future efforts for multi-centric biomedical studies. In recent work, it has been established by privacy law experts that data processed using MHE can be considered "anonymous" data under the GDPR[12]. Anonymous data, which refers to data that require unreasonable efforts to re-identify the source individuals, lies outside the jurisdiction of GDPR. Therefore, our approach has the potential to significantly simplify the requirements for contractual agreements and the obligations of data controllers with respect to regulations, such as GDPR, that often hinder multicentric medical studies. In

contrast, existing FA solutions, where the intermediate results are openly shared, present more complicated paths toward compliance, as intermediate results could still be considered personal data[6,13,14].

In cases where the potential leakage of privacy in the final output of the federated analysis is a concern, diffP techniques can be easily incorporated into FAMHE by adding a small perturbation to the final results before they are revealed. In contrast to the conventional federated-learning approach, which requires each DP to perturb its local results before aggregating them with other parties, FAMHE enables the DPs to keep the local results encrypted and reveals only the final aggregated results. Therefore, FAMHE can use a smaller amount of added noise and achieve the same level of privacy[38]. Notably, the choices of diffP parameters suitable for analyses with a high-dimensional output, such as GWAS, can be challenging and needs to be further explored.

There are several directions in which our work could be extended to facilitate the adoption of FAMHE. Although we reproduced published studies by distributing a pooled dataset across a group of DPs, jointly analyzing multiple datasets by using FAMHE that could not be combined otherwise would be a challenging yet important milestone for this endeavor. Our work demonstrates FAMHE's applicability on a reliable baseline and constitutes an important and necessary step towards building trust in our technology and fostering its adoption, thus enabling its use for the discovery of new scientific insights. Furthermore, we will extend the capabilities of FAMHE by developing additional protocols for a broader range of standard analysis tools and ML algorithms in biomedical research (e.g., proportional-hazard regression models). A key step in this direction is to make our implementation of FAMHE easily configurable by practitioners for their own applications. Specifically, connecting FAMHE to existing user-friendly platforms such as MedCo[39] to make it

widely available would help empower the increasing efforts to launch multicentric medical studies and accelerate scientific discoveries.

## Methods

Here, we describe FAMHE's system and threat model, before detailing the execution of the privacy-preserving pipelines for survival curves and GWAS studies. Finally, we detail our experimental settings and explain how diffP can be ensured on the final result in FAMHE.

### System and Threat Model

FAMHE supports a network of mutually distrustful medical institutions that act as DPs and hold subjects' records. An authorized querier (see Fig. 1) can run queries, without threatening the data confidentiality and subjects' privacy. The DPs and the querier are assumed to follow the protocol and to provide correct inputs. All-but-one DPs can be dishonest, i.e., they can try to infer information about other DPs by using the protocol's outputs. We assume that the DPs are available during the complete execution of a computation. However, to account for unresponsive DPs, FAMHE can use a threshold-encryption scheme, where the DPs secret-share[40] their secret keys, thus enabling a subset of the DPs to perform the cryptographic interactive protocols.

FAMHE can be extended to withstand malicious behaviors. A malicious DP can try to disrupt the federated collaboration process, i.e., by performing wrong computations or inputting wrong results. This can be partially mitigated by requiring the DPs to publish transcripts of their computations and to produce zero-knowledge proofs of range[41], thus constraining the DPs' possible inputs. Also, the querier can try to infer information about a DP's local data from the final result. FAMHE can mitigate this inference attack by limiting the number of requests that a querier can perform and by adding noise to the final result (see "Discussion") to achieve diffP guarantees. Learning how to select the privacy parameters and to design a generic solution to apply these techniques for the wide range of applications enabled by FAMHE is part of future work.

### FAMHE's optimization techniques

Here, we describe the main optimization techniques introduced in FAMHE. We then explain how these optimizations are used in FAMHE to compute survival curves and GWAS.

In order to parallelize and efficiently perform computationally intensive tasks, we rely on the SIMD property of the underlying cryptographic scheme and on edge computing, i.e., the computations are pushed to the DPs. In MHE, a ciphertext encrypts a vector of $N$ values, and any operation (i.e., addition, multiplication, and rotation) performed on the ciphertext is executed on all the values simultaneously, i.e., SIMD. After a certain number of operations, the ciphertext needs to be refreshed, i.e., bootstrapped. A rotation is, in terms of computation complexity, one order of magnitude more expensive than an addition/multiplication, and a bootstrapping in a centralized setting is multiple orders of magnitudes (2–4) more expensive than any other operation. As the security parameters determine how many operations can be performed before a ciphertext needs to be bootstrapped, conservative parameters that incur large ciphertexts, but enable more operations without bootstrap are usually required in centralized settings. This results in higher communication and computation costs. With MHE, a ciphertext can be refreshed by a lightweight interactive protocol that, besides its efficiency, also alleviates the constraints on the cryptographic parameters and enables FAMHE to ensure a high level of security and still use smaller ciphertexts. For example, we show in Fig. 2b how FAMHE's execution time to compute a survival curve increases when doubling the size of a ciphertext (from 4096 to 8192 slots).

As discussed in the privacy-preserving pipeline for GWAS, in the case of GWAS, FAMHE efficiently performs multiple subsequent large-dimension matrix operations (Supplementary Fig. 2) by optimizing the data packing (Supplementary Fig. 3) to perform several multiplications in parallel and to minimize the number of transformations required on the ciphertexts. FAMHE builds on the DPs' ability to compute on their cleartext local data and combine them with encrypted data, thus reducing the overall computation complexity. GWAS also requires non-polynomial functions, e.g., the inverse of a matrix, to be evaluated on ciphertexts, which is not directly applicable in HE. In FAMHE, these non-polynomial functions are efficiently approximated by relying on Chebyshev polynomials. We chose to rely on Chebyshev polynomials instead of on least-square polynomial approximations in order to minimize the maximum approximation error hence avoid that the function diverges on specific inputs. This technique has been shown to accurately approximate non-polynomial functions in the training of generalized models[24] and neural networks[42], which further shows the generality and applicability of our proposed framework.

FAMHE combines the aforementioned features to efficiently perform FA with encrypted data. In GWAS, for example, we rely on the Gauss–Jordan (GJ) method[43] to compute the inverse of the covariance matrix. We chose this algorithm as it can be efficiently executed by relying on the aforementioned features: row operations can be efficiently parallelized with SIMD and divisions are replaced by polynomial approximations.

### Privacy-preserving pipeline for survival curves

Survival curves are generally estimated with the Kaplan–Meier estimator[44]

$$\hat{S}(t) = \prod_{j:\, t_j \leq T} \left( 1 - \frac{d_j}{n_j} \right), \tag{1}$$

where $t_j$ is a time when at least one event has occurred, $d_j$ is the number of events at time $t_j$, and $n_j$ is the number of individuals known to have survived (or at risk) just before the time point $t_j$. We show in Fig. 2a the exact replica of the survival curve presented by Samstein et al.[30] produced by our distributed and privacy-preserving computation. In a survival curve, each step down is the occurrence of an event. The ticks indicate the presence of censored patients, i.e., patients who withdrew from the study. The number of censored patients at time $t_j$ is indicated by $c_j$. As shown in Supplementary Fig. 1, to compute this curve, each DP $i$ locally computes, encodes, and encrypts a vector of the form $n_0^{(i)}, c_0^{(i)}, d_0^{(i)}, ..., n_T^{(i)}, c_T^{(i)}, d_T^{(i)}$ containing the values $n_j^{(i)}, c_j^{(i)}, d_j^{(i)}$ corresponding to each time point $t_j$ for $t_j = 0, ..., T$. All the DPs' vectors are then collectively aggregated. The encryption of the final result is then collectively switched from the collective public key to the querier's public key that can decrypt the result with its secret key and generate the curve following Eq. (1).

### Privacy-preserving pipeline for GWAS

We briefly describe the genome-wide association-study workflow before explaining how we perform it in a federated and privacy-preserving manner. We conclude by detailing how we obtained our baseline GWAS results in "Results" with the PLINK software.

We consider a dataset of $p$ samples, i.e., patients. Each patient is described by $f$ features or covariates (with indexes 1 to $f$). We list all recurrent symbols and acronyms in Supplementary Table 3. Hence, we have a covariates matrix $\mathbf{X} \in \mathbb{R}^{(p \times f)}$. Each patient also has a phenotype or label, i.e., $\mathbf{y} \in \mathbb{R}^{(p \times 1)}$ and $v$ variant values, i.e., one for each variant considered in the association test. The $v$ variant values for all $p$ patients form another matrix $\mathbf{V} \in \mathbb{R}^{(p \times v)}$. To perform the GWAS, for each variant $i$, the matrix $\mathbf{X}' = [\mathbf{I}, \mathbf{X}, \mathbf{V}[:, i]] \in \mathbb{R}^{(p \times (f+2))}$, i.e., the matrix $\mathbf{X}$ is augmented by a column of 1s (intercept) and the column of one variant $i$, is constructed. The vector $\mathbf{w} \in \mathbb{R}^{(f+2)}$ is then obtained by $\mathbf{w} = (\mathbf{X}'^T \mathbf{X}')^{(-1)} \mathbf{X}'^T \mathbf{y}$. The $P$ value for variant $i$ is then obtained with

$$P = 2 \cdot \text{pnorm}\left( -\left| \frac{\mathbf{w}[f+2]}{\sqrt{\left(\text{MSE}(\mathbf{y}, \mathbf{y}') \cdot (\mathbf{X}'^T \mathbf{X}')^{(-1)}[f+2; f+2]\right)}} \right| \right),$$

where pnorm is the cumulative distribution function of the standard normal distribution, $\mathbf{w}[f+2]$ is the weight corresponding to the variant, $\text{MSE}(\mathbf{y}, \mathbf{y}')$ is the mean-squared error obtained from the prediction $\mathbf{y}'$ computed with $\mathbf{w}$, and $(\mathbf{X}'^T \mathbf{X}')^{(-1)}[f+2; f+2]$ corresponds to the standard error of the variant weight.

Although this computation has to be performed for each variant $i$, we remark that $\mathbf{X}$ is common to all variants. In order to compute $(\mathbf{X}^T \mathbf{X})^{(-1)}$ only once before adjusting it for each variant and thus obtain $(\mathbf{X}'^T \mathbf{X}')^{(-1)}$, we rely on the Shermann–Morrison formula[45] and the method presented in the report on cryptographic and privacy-preserving primitives (p. 52) of the WITDOM European project[46]. We describe this approach, i.e., FAMHE-GWAS, in Supplementary Fig. 2. Each DP$_i$ has a subset of $p_i$ patients. For efficiency, the DPs are organized in a tree structure and one DP is chosen as the root of the tree DP$_R$. We remark that, as any exchanged information is collectively encrypted, this does not have any security implications. In a CA, each DP encrypts (E()) its local result with the collective key, aggregates its children DPs encrypted results with its encrypted local results, and sends the sum to its parent DP such that DP$_R$ obtains the encrypted result aggregated among all DPs. We recall here that with the homomorphic-encryption scheme used, vectors of values can be encrypted in one ciphertext and that any operation performed on a ciphertext is simultaneously performed on all vector elements, i.e., SIMD. We rely on this property to parallelize the operations at multiple levels: among the DPs, among the threads in each DP and among the values in the ciphertexts.

We rely on the GJ method[43] to compute the inverse of the encrypted covariance matrix. We chose this algorithm as it requires only row operations, which can be efficiently performed with SIMD. The only operation that is not directly applicable in HE is the division that we approximated with a Chebyshev polynomial. Note that we avoid any other division in the protocol by pushing them to the last step that is executed by the querier $Q$ after decryption. In Supplementary Fig. 2, we keep $1/c$ until decryption.

In Supplementary Fig. 2, we describe how we further reduce the computation overhead by obtaining the covariates' weights $\mathbf{w}'$ with a lightweight federated gradient descent (FGD), by reporting the obtained covariates' contributions in the phenotype $y$, which becomes $y''$. To compute the $P$ value, we then compute only one element of the covariance inverse matrix $(\mathbf{X}'^T \mathbf{X}')^{(-1)}[f+2; f+2]$, instead of the entire inverse. To perform the FGD, we follow the method described by Froelicher et al.[24], without disclosing any intermediate values.

We describe in Supplementary Fig. 3 how the (main) values used in both protocols are packed to optimize the communication and the number of required operations (multiplications, rotations). We perform permutations, duplications,

and rotations on cleartext data that are held by the DPs (indicated in orange in Supplementary Figure 3); and we avoid, as much as possible, the operations on encrypted vectors. Note that rotations on ciphertexts are almost one order of magnitude slower than multiplications or additions and should be avoided when possible. As ciphertexts have to be aggregated among DPs, a tradeoff has to be found between computation cost (e.g., rotations) and data packing, as a smaller packing density would require the exchange of more ciphertexts.

In both protocols, all operations for $v$ variants are executed in parallel, due to the ciphertext packing (SIMD). For a 128-bit security level, the computations are performed simultaneously for 512 variants with FAMHE-GWAS and for 8192 with FAMHE-FastGWAS. These operations are further parallelized due to multithreading and to the distribution of the workload among the DPs. We highlight (in bold) the main steps and aggregated values in the protocol and note that DPs' local data are in cleartext, whereas all exchanged data are collectively encrypted (E()).

**Baseline computations with PLINK**. As explained in the "Results" section, we relied on the PLINK software to obtain our baseline results for the (i) Original approach in which GWAS is computed on the entire centralized dataset, (ii) the Independent approach in which each DP performs the GWAS on its own subset of the data, and (iii) for the Meta-analysis in which the DPs perform the GWAS on their local data before combining their results. For (i) and (ii), we relied on PLINK 2.0 and its linear regression (–glm option)-based association test. For (iii), we relied on PLINK 1.9 and used the weighted-Z test approach to perform the meta-analysis.

**Experimental settings**. We implemented our solutions by building on top of Lattigo[33], an open-source Go library for lattice-based cryptography, and Onet[47], an open-source Go library for building decentralized systems. The communication between DPs is done through TCP, with secure channels (by using TLS). We evaluate our prototype on an emulated realistic network, with a bandwidth of 1 Gbps and a delay of 20 ms between every two nodes. We deploy our solution on 12 Linux machines with Intel Xeon E5-2680 v3 CPUs running at 2.5 GHz with 24 threads on 12 cores and 256 GB of RAM, on which we evenly distribute the DPs. We choose security parameters to always achieve a security level of 128 bits.

**Differentially private mechanism**. DiffP is a privacy-preserving approach, introduced by Dwork[26], for reporting results on statistical datasets. This approach guarantees that a given randomized statistic, $\mathcal{M}(DS) = R$, computed on a dataset DS, behaves similarly when computed on a neighbor dataset DS′ that differs from DS in exactly one element. More formally, $(\epsilon, \delta)$-diffP[48] is defined by $\Pr[\mathcal{M}(DS) = R] \leq \exp(\epsilon) \cdot \Pr[\mathcal{M}(DS') = R] + \delta$, where $\epsilon$ and $\delta$ are privacy parameters: the closer to 0 they are, the higher the privacy level is. $(\epsilon, \delta)$-diffP is often achieved by adding noise to the output of a function $f(DS)$. This noise can be drawn from the Laplace distribution with mean 0 and scale $\frac{\Delta f}{\epsilon}$, where $\Delta f$, the sensitivity of the original real-valued function $f$, is defined by $\Delta f = \max_{D,D'} ||f(DS) - f(DS')||_1$. Other mechanisms, e.g., relying on a Gaussian distribution, were also proposed[26,49].

As explained before, FAMHE can enable the participants to agree on a privacy level by choosing whether to yield exact or obfuscated, i.e., differentially private results, to the querier. We also note that our solution would then enable the obfuscation of only the final result, i.e., the noise can be added before the final decryption, and all the previous steps can be executed with exact values as no intermediate value is decrypted. This is a notable improvement with respect to existing federated-learning solutions, based on diffP[38], in which the noise has to be added by each DP at each iteration of the training. In the solution by Kim et al.[38], each DP perturbs its locally computed gradient such that the aggregated perturbation, obtained when the DPs aggregate (combine) their locally updated model, is $\epsilon$-differentially private. This is achieved by having each DP generate and add a partial noise such that, when aggregated, the total noise follows the Laplace distribution. The noise magnitude is determined by the sensitivity of the computed function and this sensitivity is similar for each DP output and for the aggregated final result. This means that, as the intermediate values remain encrypted in FAMHE, a noise with the same magnitude can be added only once on the final result, thus ensuring the same level of privacy with a lower distortion of the result.

**Reporting summary**. Further information on research design is available in the Nature Research Reporting Summary linked to this article.

## Data availability

We replicated two existing medical studies, Samstein et al.[30] and McLaren et al.[31]. The original data used by Samstein et al. and in our work is available at http://www.cbioportal.org/study/summary?id=tmb_mskcc_2018. The data used by McLaren et al. and in our work are protected and under the responsibility of the authors of the original study. Interested researchers should contact these authors if they wish to access the dataset.

## Code availability

Our solution partially relies on open-source software and public libraries (i.e., the cryptographic library Lattigo[33] and the decentralized systems library Onet[47]). Our code is currently not publicly available as its license does not allow for open-source redistribution. Pseudocode of the used algorithms and protocols is provided for completeness in the "Methods" section Upon request sent to the corresponding author(s), we can provide binaries that, in combination with open-source resources, can be used for the sole purpose of verifying and reproducing the experiments in the manuscript.

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

## Acknowledgements

We would like to thank Apostolos Pyrgelis for providing valuable feedback and the DeDiS lab, led by Bryan Ford, for providing software support. This work was partially supported by grant #2017-201 of the Strategic Focal Area "Personalized Health and Related Technologies (PHRT)" of the ETH Domain. This work was also partially supported by NIH R01 HG010959 (to B.B.) and NIH DP5 OD029574 (to H.C.).

## Author contributions

J.R.T.-P., J.L.R., and J.-P.H. conceived the study. D.F. and J.R.T.-P. developed the methods. D.F. and J.S.S. implemented the software and performed experiments, the results of which were validated by M.A.C. and J.F. H.C. and B.B. provided biological interpretation of the results and helped revise the manuscript. All authors contributed to the methodology and wrote the manuscript.

## Competing interests

J.R.T.-P. and J.-P.H. are co-founders of the start-up Tune Insight (https://tuneinsight.com). All authors declare no other competing interests.
