## [Peer Review File · Nature Communications]

Reviewers' Comments:

Reviewer #1:

Remarks to the Author:

This paper presents a privacy preserving federated analytics framework for medical data analysis. Specifically, the authors proposed a multi-party homomorphic encryption (FAMHE) framework which consists of steps including query, iterative local computation and encryption, collective aggregation, and finally collective decryption. Empirical results on two artificially constructed examples are presented to demonstrate the effectiveness of FAMHE. Overall the paper is well-written, but I have the following concerns.

1. This paper mixed up federated learning and distributed homomorphic encryption. One key advantage of federated learning is it supports complicated model training process (such as the ones need complicated gradient descent or back propagation), while HE can only handle simple primitive calculations. As the paper has shown, in both empirical examples only simple calculations involving counts and matrix plus/multiplication are involved. It is not clear if HE can go beyond that, e.g., if the authors can demonstrate an example with deep learning or random forest model that would be great. Actually in secure federated analytics there are two main modules, one is federated analytics where you can learn complicated models in a distributed way without accessing local data simultaneously, the other is security, where you protect the intermediate model parameter estimations when transmitting them to the central server (in centralized setting) or with neighbor clients (in decentralized setting). For example we can add differential privacy in the second step. It seems FAMHE proposed in this paper only has the second step but how to apply it on complicated model learning is not clear at all.

2. The authors claimed that "To demonstrate the readiness and maturity of our approach to handle real-world use cases for multicentric medical studies, as opposed to previous published attempts that focused mainly on artificial demonstrative examples". However, in the empirical study the authors still artificially partition them into multiple pieces and validate the proposed mechanism. It would be important for the authors to validate their proposed approach in a real multi-party setting and discuss/demonstrate the benefits.

Reviewer #2:

Remarks to the Author:

The proposed solution consists of two cryptographic tools (multiparty homomorphic encryption, differential privacy). Technical originality is mainly contained in the computation of privacy-preserving GWAS. Three techniques are leveraged for the privacy-preserving protocol tailored to GWAS over multiple domains. (1) It requires solving a large number of linear regression models where most of the covariates are commonly shared. The proposed protocol reduces computation by leveraging the Sherman-Morrison formula. (2) multiparty versions of homomorphic encryption/key switching (BFV and CKKS) are used to attain privacy-preserving computation over multiple data providers. (3) For speeding up computation, SIMD-like parallelism using the characteristics of RLWE is leveraged.

The first technique is a well-known idea while introducing this technique into homomorphic encryption seems new.

The second technique is a simple combination of existing ideas into the proposed framework.

The third technique is widely used for computation with homomorphic encryption based on RLWE. Overall, the technical originality introduced by the proposed framework seems to fairly weak.

Table 1 compares the proposed framework to existing methods. Some criterion needs clarification. Dis-honest majority:

The proposed framework is secure against the dis-honest majority at the decryption phase, while misbehavior of even a single party in the middle of the protocol execution would cause wrong results. In this sense, the proposed protocol is not secure against the dishonest majority. In this sense, is the companion concerning this aspect is fair? This point should be stated more clearly.

Computation flexibility:

Since the proposed framework is basically implemented with homomorphic encryption, it would be flexibly transformed into other types of analytics in principle. However, the protocol is tailored to GWAS (and Kaplan-Meier) for computational efficiency, and such transformation with preserving computational efficiency is not straightforward. Considering that secret-sharing-based SMC is also applicable to any kind of computation, a comparison concerning this aspect should be discussed more carefully.

Application to real cases:

The reviewer understands that the presented experiments are conducted with the same data as used in real biomedical studies. However, if the focus is on the computational capability of the proposed protocol, whether or not the data is used in real cases might not be significant. If the experiment is conducted to find a new scientific discovery, it would be a clear differentiator.

The followings are comments to improve the manuscript.

In Fig 4, (b) and (c), the results of FAMHE-fast and FAMHE are not comparable. It would be better to be comparable.

Also, evaluation of communication bandwidth in the same setting would help readers to estimate the overall execution time.

Reviewer #3:

Remarks to the Author:

This paper presents cryptographic protocols for secure genome analysis. The framework proposed by the paper is general, and the paper focuses on two primary applications: Kaplan-Meier survival analysis, and genome-wide association studies (GWAS). In both cases, they successfully use their framework to replicate the result of prior scientific studies (the study of Samstein et al. in the case of Kaplan-Meier survival analysis and that of McLaren et al. in the case of GWAS).

In the model considered in this paper, there are multiple independent data providers who want to evaluate a joint function of their inputs (this typically consists of sensitive patient data) without needing to share any intermediary results or revealing any information other than what is revealed by the output of the computation. The paper relies on a recent notion called "multiparty homomorphic encryption" (MHE) which is a hybrid of multiparty computation and homomorphic encryption. The advantage of MHE is that the computation is mostly non-interactive (i.e., the different data providers locally operate on their local version of the data); communication is only needed for synchronization and decrypting the final results. Because there is much more limited communication compared to previous techniques based on multiparty computation, the paper argues that the approach scales much better to handle more data providers or computing parties. At the same time, the use of cryptographic techniques still ensures *exact* computation of the genomic quantities of interest (in contrast to techniques like differential privacy that relies on introducing noise into the computation to hide data).

Overall, this is a nice addition to a growing list of systems for leveraging cryptographic tools for private computation on genomic data. The proposed protocols successfully replicate existing studies, which demonstrates the effectiveness of the proposed approach in several real settings.

In my view, the main weakness in this work is that it seems to be relying on an existing cryptographic protocol (namely, the multiparty homomorphic encryption scheme by Mouchet et al (ePrint 2019)) and using it to replicate existing genomic studies. While I appreciate the engineering effort to take the system and apply it to new datasets, I would also like to see some discussion of technical insights beyond the combination of existing tools. For example, are there specific modifications that were needed on the cryptography side in order to apply it to real datasets? Are there compromises made on the genomic computation side to make the computation more suitable for computation using the proposed framework? In the case of the latter, the paper should provide a description of the types of computations that are suitable for the MHE framework, and how the specific scenarios envisioned by this paper take advantage of that. The paper would be much stronger if it could provide some discussion of technical challenges that

needed to be overcome in order to support the functionalities described in this work.

The paper's introduction motivates the design of the proposed work by discussing a number of alternative cryptographic approaches (which have been used in a number of previous systems) and discussing their limitations. However, I do not see any *concrete* comparison against the previous approaches in the current manuscript. How does the system compare against existing systems for these tasks. For instance, there have been multiple works on privacy-preserving GWAS based on differential privacy, multiparty computation, and homomorphic encryption. How does the proposed techniques compare against the existing systems (in terms of end-to-end computation time, communication requirements, overall cost, etc.)? What is the primary improvement in the current system compared to previous ones? Are there any limitations of the current system compared to previous ones? Even providing a rough ballpark estimate of the costs of competing solutions would provide better context for understanding the main improvements of the system over existing work.

Since the main contribution seem to be on the systems design and the implementation, I would also like to see a more comprehensive systems evaluation: for example, measurements of the total communication between parties, the number of rounds, etc. (and how these compare with previous works). What are the bottlenecks in the system? Is it computation or communication? How would things change with a different network configuration? In the current setup, it seems like the data providers are essentially co-located in a LAN configuration; how realistic is it that all data providers are co-located? What would the protocol costs look like if the providers are operating over a wide area network?

On the cryptography side, what are the cryptographic assumptions that are needed? Since homomorphic encryption is used, I assume that this would require stronger cryptographic assumptions compared to previous systems? This is fine, but should be stated clearly in the paper.

There is also some discussion of using the CKKS approximate FHE scheme. If my understanding is correct, there was an attack on this scheme even in the presence of passive adversaries (Li-Micciancio; to appear at EUROCRYPT 2021; preprint available on ePrint 2020). Does this affect the claimed security guarantees? Are there specific mitigations that are deployed here? When the paper describes an "adaptation" of CKKS, what does this refer to? It sounds like there are some proposed modifications to the underlying FHE scheme. If this is the case, then the modifications should be clearly described (e.g., in the Supplementary Material) and a concrete security proof should be provided. Otherwise, it is difficult to assess the formal security guarantees the system provides.

To summarize, this paper has nice contributions to the study of privacy-preserving genomic computations. However, I have some questions regarding the novelty of the techniques and on how the results compare against previous cryptographic systems for these same types of computations. Clarifying both aspects would strengthen the paper and better highlight the primary technical contributions of the work.

Response to Reviewers' Comments for Manuscript NCOMMS-21-09899-T
Entitled: "Truly Privacy-Preserving Federated Analytics for Precision Medicine with Multiparty Homomorphic Encryption"

We thank the reviewers for their insightful and constructive comments that we have carefully reviewed and addressed in our revised submission. We have made the following major changes to the manuscript:

1. **Clarification of our technical contributions.** As suggested by all the reviewers, we have clarified the technical contributions of our work by expanding the descriptions of our methodology in the Introduction, Results, and Discussion sections. We now emphasize that our work builds upon the framework of MHE for designing efficient federated workflows for essential biomedical tasks, such as survival analyses and GWAS. Our success in these tasks is enabled by the optimized MHE novel routines (e.g., for matrix inversion) we newly developed in this work, which we now clarify in the text. We now also emphasize that FAMHE scales to large numbers of data providers while providing end-to-end privacy protection and without loss of accuracy. The combination of these features distinguishes FAMHE from existing approaches for federated analytics. We have added detailed comparisons with existing methods in Online Methods and Supplementary Notes 1 & 4.
2. **Expanded discussion of the biomedical implications of our results.** In response to Reviewers #1 and #2's comments, we revamped our Results section to more clearly communicate the implications of our results to the biomedical community. We believe our successful replications of existing peer-reviewed studies by using FAMHE clearly demonstrate its potential to enable similar studies in the future in a *federated secure* manner. Moreover, we now include an additional baseline result in our GWAS analysis that is based on each data provider independently performing GWAS on its own subset of data; our results show that the data providers are not able to solely obtain results that are as scientifically informative as those of the pooled or our federated analyses, thus demonstrating further benefits of federated collaboration enabled by our work. Conducting a new multi-centric study by using our approach to discover novel scientific insights is certainly an important direction that we will explore in future work. We have also clarified these points in the Introduction and Discussion sections.
3. **Additional benchmark experiments and comparisons.** Following the suggestions of Reviewers #2 and #3, we have made our evaluation more comprehensive by including additional experimental data in Results. We now provide a detailed breakdown of FAMHE execution time (including both communication and computation) and evaluate both our approaches based on FAMHE for GWAS for all performance metrics. To further illustrate the value of collaboration, we have added another baseline for GWAS, where each data holder independently performs GWAS. As suggested by Reviewer #3, we now provide in Supplementary Note 4 and discuss in Results, a concrete estimation-based comparison between FAMHE and existing cryptographic frameworks.

To increase the overall clarity of our work, we have made various textual clarifications and improvements to the figures throughout the manuscript.

We would also like to mention that two co-authors, Drs. Hyunghoon Cho (Broad Institute) and Bonnie Berger (MIT), have been added to the manuscript for their contributions to this work and involvement in the revision. All authors have agreed to this change.

We provide detailed responses to individual comments from the reviewers below (in blue).

Reviewer #1:

This paper presents a privacy preserving federated analytics framework for medical data analysis. Specifically, the authors proposed a multi-party homomorphic encryption (FAMHE) framework which consists of steps including query, iterative local computation and encryption, collective aggregation, and finally collective decryption. Empirical results on two artificially constructed examples are presented to demonstrate the effectiveness of FAMHE. Overall the paper is well-written, but I have the following concerns.

1. This paper mixed up federated learning and distributed homomorphic encryption. One key advantage of federated learning is it supports complicated model training process (such as the ones need complicated gradient descent or back propagation), while HE can only handle simple primitive calculations. As the paper has shown, in both empirical examples only simple calculations involving counts and matrix plus/multiplication are involved. It is not clear if HE can go beyond that, e.g., if the authors can demonstrate an example with deep learning or random forest model that would be great. Actually in secure federated analytics there are two main modules, one is federated analytics where you can learn complicated models in a distributed way without accessing local data simultaneously, the other is security, where you protect the intermediate model parameter estimations when transmitting them to the central server (in centralized setting) or with neighbor clients (in decentralized setting). For example we can add differential privacy in the second step. It seems FAMHE proposed in this paper only has the second step but how to apply it on complicated model learning is not clear at all.

We thank the reviewer for pointing out a potential source of confusion regarding the difference between federated learning and distributed/multiparty HE (MHE). As we described in Introduction, we view our FAMHE framework as a means to achieve the same goals as federated learning (i.e., jointly analyze distributed datasets) without relying on the insecure sharing of intermediate results such as gradients. We achieve this by sharing intermediate results only in an encrypted form under multiparty HE, which still allows local computations to be performed on both the shared encrypted and local plaintext data. The reviewer is correct in commenting that the flexibility of the local computation in FAMHE is determined by the underlying HE framework. However, our work demonstrates that, in practice, this aspect is not an impediment to supporting the federated execution of essential biomedical tasks such as survival analysis and GWAS. Importantly, our GWAS solutions newly handle complex operations, such as matrix inversion, as well as gradient computation mentioned by the reviewer. This shows that FAMHE can indeed support complex operations while providing formal security guarantees that many federated learning approaches lack.

Furthermore, we note that recent works, including from our own team, show the applicability of MHE for training complex ML models (e.g., generalized linear models and neural networks). We envision exploiting these recent techniques to extend FAMHE to other important biomedical tasks in the future. We have clarified these points in Introduction and Discussion.

2. The authors claimed that "To demonstrate the readiness and maturity of our approach to handle real-world use cases for multicentric medical studies, as opposed to previous published attempts that focused mainly on artificial demonstrative examples". However, in the empirical study the authors still artificially partition them into multiple pieces and validate the proposed mechanism. It would be important for the authors to validate their proposed approach in a real multi-party setting and discuss/demonstrate the benefits.

We appreciate this comment from the reviewer. Our intention behind the quoted text was to emphasize that our work accurately reproduces the results of *published, peer-reviewed studies*, which we believe clearly demonstrates the ability of FAMHE to support similar studies in the future in a federated setting. We sought to draw a distinction from existing work in the security and privacy literature that usually rely on synthetic datasets (or toy examples). Note that aside from the partitioning of the dataset, all other aspects of our experiments directly follow the original studies. Hence, we believe our experiments closely reflect a real application of FAMHE. Conducting a new multi-centric study by using our approach to discover novel scientific insights is certainly an important direction that we will explore in future work. To this end, we believe that our work, demonstrating FAMHE's applicability on a reliable baseline, is an important and necessary step towards building trust in our technology to foster its adoption. We have revised the text to clarify these points in Abstract, Introduction and Discussion.

To further address the reviewer's concern about the benefit of our federated analysis approach in a realistic scenario, we now include an additional baseline result in our GWAS analysis that is based on each data provider independently performing GWAS on its own subset of data. As depicted in revised Figure 3 (and Supplementary Figure S4), our results show that the data providers are not able to solely obtain results that are as scientifically informative as those of the pooled or our federated analyses; in fact, these independent analyses typically did not result in any statistically significant finding due to lack of power. Thus we further demonstrate the benefits of federated collaboration enabled by our work.

Reviewer #2:

1. The proposed solution consists of two cryptographic tools (multiparty homomorphic encryption, differential privacy). Technical originality is mainly contained in the computation of privacy-preserving GWAS. Three techniques are leveraged for the privacy-preserving protocol tailored to GWAS over multiple domains. (1) It requires solving a large number of linear regression models where most of the covariates are commonly shared. The proposed protocol reduces computation by leveraging the Sherman-Morrison formula. (2) multiparty versions of homomorphic encryption/key switching (BFV and CKKS) are used to attain privacy-preserving computation over multiple data providers. (3) For speeding up computation, SIMD-like parallelism using the characteristics of RLWE is leveraged. The first technique is a well-known idea while introducing this technique into homomorphic encryption seems new. The second technique is a simple combination of existing ideas into the proposed framework. The third technique is widely used for computation with homomorphic encryption based on RLWE. Overall, the technical originality introduced by the

proposed framework seems to fairly weak.

We believe that the main contribution of our work lies in introducing the combination of these building block techniques, correctly identified by the reviewer, as a methodological solution for federated analytics/learning in biomedical research and in demonstrating the effectiveness of our approach in two essential tasks of broad interest: Kaplan-Meier survival analysis and GWAS. Although each of the techniques has been independently considered in other domains, leveraging them jointly and adapting them to design practical protocols for survival analysis and GWAS is novel and represents significant progress in the field. In the manuscript, we describe how we combine ideas from HE, interactive protocols, and edge-computing by the data providers in a task-specific manner to obtain efficient solutions that scale similarly to insecure federated solutions, but with full privacy protection. We now clarify our conceptual and technical contributions throughout the manuscript. We also have expanded our comparison with existing solutions (in Discussion, Online Methods and Supplementary Notes 1 & 4), thus highlighting our advances with respect to existing approaches.

2. Table 1 compares the proposed framework to existing methods. Some criterion needs clarification. Dis-honest majority: The proposed framework is secure against the dis-honest majority at the decryption phase, while misbehavior of even a single party in the middle of the protocol execution would cause wrong results. In this sense, the proposed protocol is not secure against the dishonest majority. In this sense, is the companion concerning this aspect is fair? This point should be stated more clearly.

We thank the reviewer for pointing out this potentially confusing statement. In our threat model, we consider passive adversaries who do not interfere with the protocol but can collude and try to infer information from their combined information. Under this setting, our framework remains secure even when all-but-one data providers are dishonest. We have updated the text to clarify this point in Introduction, Table 1 and Online Methods (System & Threat Model). In the latter, we also added a discussion about how our framework can be extended to withstand malicious behavior that the reviewer alluded to.

3. Computation flexibility: Since the proposed framework is basically implemented with homomorphic encryption, it would be flexibly transformed into other types of analytics in principle. However, the protocol is tailored to GWAS (and Kaplan-Meier) for computational efficiency, and such transformation with preserving computational efficiency is not straightforward. Considering that secret-sharing-based SMC is also applicable to any kind of computation, a comparison concerning this aspect should be discussed more carefully.

The reviewer insightfully pointed out that FAMHE is flexible in terms of the range of computations it can support. Indeed, our work demonstrates that FAMHE can be efficiently used for both simple (i.e., survival curves) and complex (i.e., GWAS involving millions of regression models and matrix inversion) analytic workflows that are essential in biomedical research. As the reviewer mentioned, tailoring these protocols for the target applications requires careful design and optimization efforts; **this is a key contribution of our work.** Other instantiations of our framework, e.g., for training more sophisticated machine-learning

models, are certainly feasible, and we plan to develop them in future work. We now add this point to the Discussion. Note that recent works [F,G,H], including from our own team, have shown the applicability of MHE for training complex ML models (e.g., generalized linear models and neural networks).

As pointed out by the reviewer, secret sharing-based SMC solutions benefit from similar computational flexibility to our framework. However, we believe that the framework of MHE offers a number of key benefits over SMC; this motivated our development of FAMHE solutions. Contrary to SMC-based approaches, FAMHE scales efficiently to large numbers of data providers. This enables data providers to keep their data locally, instead of outsourcing them to a limited number of computing parties via secret sharing. This aspect of FAMHE further allows the data providers to compute directly on their local cleartext data and combine them with encrypted data, i.e., edge-computing, which makes the overall workflow more efficient. We now highlight these points in Introduction and Discussion. We also have included a discussion on how FAMHE's performance compares with SMC-based approaches in Online Methods and Supplementary Notes 1 & 4.

4. The reviewer understands that the presented experiments are conducted with the same data as used in real biomedical studies. However, if the focus is on the computational capability of the proposed protocol, whether or not the data is used in real cases might not be significant. If the experiment is conducted to find a new scientific discovery, it would be a clear differentiator.

Although we agree with the reviewer that using FAMHE to conduct a new scientific study across multiple institutions would be immensely valuable, we emphasize that the focus of our work is to demonstrate the practical feasibility of our FAMHE-based solutions, rather than to discover new scientific insights. We believe that this is a necessary key step towards building trust in our technology and fostering its adoption, thus enabling its use for new scientific discoveries. The use of datasets from real centralized studies is critically important to our work, as it enables us to compare the results of our federated experiments with the published findings to see whether the original studies could have made the same scientific discoveries if the analysis was performed using our federated approach. Our results show that, indeed, FAMHE can accurately and efficiently reproduce published findings while keeping the data distributed across parties. We believe this clearly demonstrates the ability of our solutions to enable similar studies in the future and in a federated setting. In future work, we will collaborate with biomedical researchers to launch new federated studies by using the methods introduced in this work. As described in Discussion, this effort would further require integrating our solutions into a user-friendly platform that can be used by practitioners. We appreciate the reviewer for recognizing this important future direction. In response to this comment and to clarify the main goal of this work and the implications of our results, we made several textual clarifications throughout Introduction, Results, and Discussion.

5. In Fig 4, (b) and (c), the results of FAMHE-fast and FAMHE are not comparable. It would be better to be comparable. Also, evaluation of communication bandwidth in the same setting would help readers to estimate the overall execution time.

We thank Reviewer #2 for these helpful suggestions. We have incorporated the suggested improvements into our manuscript. Figure 4 now includes the execution times of both

FAMHE-GWAS and FAMHE-FastGWAS in all graphs. We also indicated in each graph the portion of FAMHE execution time dedicated to communication. Furthermore, in Figure 4a, we now show how the FAMHE execution time would change if data providers communicate over a wide-area network (WAN), with half the available bandwidth, and with double the communication delay. These updated results show that FAMHE remains practical in a WAN setting, and they further support the practical feasibility of our FAMHE solutions.

Reviewer #3:

This paper presents cryptographic protocols for secure genome analysis. The framework proposed by the paper is general, and the paper focuses on two primary applications: Kaplan-Meier survival analysis, and genome-wide association studies (GWAS). In both cases, they successfully use their framework to replicate the result of prior scientific studies (the study of Samstein et al. in the case of Kaplan-Meier survival analysis and that of McLaren et al. in the case of GWAS).

In the model considered in this paper, there are multiple independent data providers who want to evaluate a joint function of their inputs (this typically consists of sensitive patient data) without needing to share any intermediary results or revealing any information other than what is revealed by the output of the computation. The paper relies on a recent notion called "multiparty homomorphic encryption" (MHE) which is a hybrid of multiparty computation and homomorphic encryption. The advantage of MHE is that the computation is mostly non-interactive (i.e., the different data providers locally operate on their local version of the data); communication is only needed for synchronization and decrypting the final results. Because there is much more limited communication compared to previous techniques based on multiparty computation, the paper argues that the approach scales much better to handle more data providers or computing parties. At the same time, the use of cryptographic techniques still ensures *exact* computation of the genomic quantities of interest (in contrast to techniques like differential privacy that relies on introducing noise into the computation to hide data).

Overall, this is a nice addition to a growing list of systems for leveraging cryptographic tools for private computation on genomic data. The proposed protocols successfully replicate existing studies, which demonstrates the effectiveness of the proposed approach in several real settings.

We thank the reviewer for their excellent summary, kind remarks and recognition of the value of our work.

1. In my view, the main weakness in this work is that it seems to be relying on an existing cryptographic protocol (namely, the multiparty homomorphic encryption scheme by Mouchet et al (ePrint 2019)) and using it to replicate existing genomic studies. While I appreciate the engineering effort to take the system and apply it to new datasets, I would also like to see some discussion of technical insights beyond the combination of existing tools. For example, are there specific modifications that were needed on the cryptography side in order to apply it to real datasets? Are there compromises made on the genomic computation side to make the computation more suitable for computation using the proposed framework? In the case of the latter, the

paper should provide a description of the types of computations that are suitable for the MHE framework, and how the specific scenarios envisioned by this paper take advantage of that. The paper would be much stronger if it could provide some discussion of technical challenges that needed to be overcome in order to support the functionalities described in this work.

We appreciate the reviewer's thoughtful and constructive comment. We agree that the technical contributions of our work were not sufficiently discussed in the initial manuscript. To address this concern, we have extensively edited the text in our revised manuscript.

We now explain in more detail how, in order to efficiently perform federated analytics with encrypted data, FAMHE builds on its multiparty setting and on MHE features. We notably designed secure federated workflows for survival curves and GWAS that optimize the use of local cleartext data in conjunction with shared encrypted data, i.e., edge computing. This enables us to optimize the balance between operations on cleartext and encrypted data, thus optimizing FAMHE's entire workflow. We also rely on FAMHE distributed settings to replace costly cryptographic operations, e.g., refreshing or bootstrapping a ciphertext, by lightweight interactive protocols, and to use less conservative and more efficient cryptographic parameters for a high security level. This flexibility enables us to choose a ciphertext size depending on the input dimension, e.g., time points in survival curves, and the size of a matrix row in GWAS, and that provides the best possible tradeoff between parallelization and communication/computation costs. As FAMHE has to operate on large-dimensional input, we optimized its workflow to vectorize (i.e., pack values in vectors) data before encryption; we heavily rely on the Single Instruction, Multiple Data (SIMD) property of the cryptoscheme to parallelize the operations on encrypted data in a task-specific manner. Importantly, we describe how non-polynomial functions, e.g., division in matrix inversion, are approximated in FAMHE to be efficiently executed in HE. We have now clarified these points in the Results and Online Methods.

2. The paper's introduction motivates the design of the proposed work by discussing a number of alternative cryptographic approaches (which have been used in a number of previous systems) and discussing their limitations. However, I do not see any *concrete* comparison against the previous approaches in the current manuscript. How does the system compare against existing systems for these tasks. For instance, there have been multiple works on privacy-preserving GWAS based on differential privacy, multiparty computation, and homomorphic encryption. How does the proposed techniques compare against the existing systems (in terms of end-to-end computation time, communication requirements, overall cost, etc.)? What is the primary improvement in the current system compared to previous ones? Are there any limitations of the current system compared to previous ones? Even providing a rough ballpark estimate of the costs of competing solutions would provide better context for understanding the main improvements of the system over existing work.

We thank the reviewer for pointing out this omitted discussion in our manuscript. FAMHE scales similarly as non-secure or differential-privacy-based federated analytics solutions with an overhead brought by the computations on encrypted data. Computing on encrypted data allows the data provider to collectively perform analytics by exchanging, but not revealing, information among themselves. The overhead typically brought by HE is maintained

between 1 and 2 orders of magnitude due to an efficient packing, i.e., SIMD, to “edge-computing” by the DPs combining their local cleartext data with encrypted data, and due to the use of lightweight interactive protocols in place of costly cryptographic operations.

Contrary to existing SMC-based approaches for FA, FAMHE efficiently scales to large numbers of data providers that do not need to outsource their data to a limited number of computing parties. We have clarified the improvements brought by FAMHE with respect to existing solutions in Introduction, Discussion, Online Methods and Supplementary Note 1. We also now provide a concrete comparison of FAMHE with existing cryptographic frameworks in Supplementary Note 4 and discuss this in Results.

3. Since the main contribution seem to be on the systems design and the implementation, I would also like to see a more comprehensive systems evaluation: for example, measurements of the total communication between parties, the number of rounds, etc. (and how these compare with previous works). What are the bottlenecks in the system? Is it computation or communication? How would things change with a different network configuration? In the current setup, it seems like the data providers are essentially co-located in a LAN configuration; how realistic is it that all data providers are co-located? What would the protocol costs look like if the providers are operating over a wide area network?

We thank the reviewer for the suggestions. We have now improved our performance evaluation by indicating the execution time dedicated to communication in all experiments displayed in Figure 4. We show in Figure 4 (a) how smaller bandwidth and longer communication delay influence FAMHE execution time, thus mimicking data providers communicating through a wide-area network (WAN). We show that the communication accounts for less than 55% of FAMHE execution time in all cases. We also have provided a concrete comparison of FAMHE with existing cryptographic frameworks in Supplementary Note 4 and discussed in Results.

4. On the cryptography side, what are the cryptographic assumptions that are needed? Since homomorphic encryption is used, I assume that this would require stronger cryptographic assumptions compared to previous systems? This is fine, but should be stated clearly in the paper.

We thank the reviewer for this comment. In Introduction, we clarified that we assume a passive adversarial model in which $N-1$ out of N data providers can be dishonest and collude among themselves. In the same section, we explained that collectively encrypted data can be decrypted only with the participation of all data providers.

In Discussion and Online Methods, we now explain how the use of a multiparty cryptographic scheme enables FAMHE to rely on less conservative security parameters for a high 128-bit security level. This enables FAMHE to use ciphertexts of a smaller size for the same security level. Notably, as now explained in Cryptographic Background in Online Methods, the security of the underlying multiparty cryptographic schemes follows from the security of the original cryptographic schemes, i.e., the hardness of the RLWE decision problem.

5. There is also some discussion of using the CKKS approximate FHE scheme. If my understanding is correct, there was an attack on this scheme even in the presence of

passive adversaries (Li-Micciancio; to appear at EUROCRYPT 2021; preprint available on ePrint 2020). Does this affect the claimed security guarantees? Are there specific mitigations that are deployed here? When the paper describes an "adaptation" of CKKS, what does this refer to? It sounds like there are some proposed modifications to the underlying FHE scheme. If this is the case, then the modifications should be clearly described (e.g., in the Supplementary Material) and a concrete security proof should be provided. Otherwise, it is difficult to assess the formal security guarantees the system provides.

The attack on CKKS mentioned by the reviewer [E] can be considered as a semi-active attack that is not specific to CKKS, but to approximate decryption-based systems. Respectfully, it has long been known (and addressed) for threshold RLWE cryptosystems [A,B]. This attack does not change the IND-CPA properties of CKKS, and it is applicable only when a party has access to an original ciphertext and to the corresponding decrypted plaintext, or to an original ciphertext and a partially decrypted ciphertext, provided that the used secret key to decrypt or partially decrypt is not known by the party. The countermeasures [B,C,D] consist of adding noise (unrelated to differential privacy) to the decrypted plaintext or to the partially decrypted ciphertext before sending it to the corresponding party, in such a way that this noise floods and masks the spurious term that contains the (processed) secret key information. The effect of these countermeasures can be measured as reducing 1-bit of precision in the decrypted results, or effectively doubling the output encryption noise. With the cryptographic parameters that we have chosen for FAHME, this noise increase is negligible with respect to the GWAS signals or the survival computations that are several orders of magnitude larger than the introduced error; hence, the latter does not affect the accuracy of the results in our work.

Regarding the comment about our modifications to FHE schemes, we first thank the reviewer for pointing out a potential source of confusion around the security and the use of the underlying FHE schemes. We now clarify that FAMHE relies on a multiparty instantiation of FHE schemes (i.e. MHE) and builds on these schemes' features (e.g., SIMD and packing) in order to optimize its secure workflow for FA. Our task-specific optimizations aside, FAMHE does not introduce changes to the MHE schemes that affect security. Moreover, the security of the MHE schemes can in turn be proven from the security of the original centralized schemes, as demonstrated by recent works [B, F] (including from our own team). We have clarified these points in our main text and in the Cryptographic Background Section in Online Methods.

[A] G. Asharov, A. Jain, A. López-Alt, E. Tromer, V. Vaikuntanathan, and D. Wichs. Multiparty computation with low communication, computation and interaction via threshold FHE. In Annual International Conference on the Theory and Applications of Cryptographic Techniques, pages 483–501. Springer, 2012.

[B] C. Mouchet, J.R. Troncoso-Pastoriza, J.P. Bossuat, and J.P. Hubaux, "Multiparty Homomorphic Encryption from Ring-Learning-With-Errors". Cryptology ePrint archive. <https://eprint.iacr.org/2020/304.pdf> (2020)

[C] L. de Castro, C. Juvekar, and V. Vaikuntanathan. "Fast Vector Oblivious Linear Evaluation from Ring Learning with Errors". Cryptology ePrint archive. <https://eprint.iacr.org/2020/685> (2020)

[D] J.H. Cheon, S. Hong, and D. Kim. "Remark on the Security of CKKS Scheme in Practice". Cryptology ePrint archive. <https://eprint.iacr.org/2020/1581> (2020)

- [E] B. Li D. Micciancio. "On the Security of Homomorphic Encryption on Approximate Numbers". Cryptology ePrint archive. <https://eprint.iacr.org/2020/1533> (2020)
- [F] D. Froelicher, J.R. Troncoso-Pastoriza, A. Pyrgelis, S. Sav, J.S. Sousa, J.P. Bossuat, and J.P. Hubaux. "Scalable Privacy-Preserving Distributed Learning". PETS (2021)
- [G] S. Sav, A. Pyrgelis, J.R. Troncoso-Pastoriza, D. Froelicher, J.P. Bossuat, J.S. Sousa and J.P. Hubaux, J. P. "POSEIDON: Privacy-Preserving Federated Neural Network Learning". arXiv preprint arXiv:2009.00349. (2020)
- [H] W. Zheng, R. A. Popa, J. E. Gonzalez and I. Stoica. "Helen: Maliciously secure cooperative learning for linear models." In 2019 IEEE Symposium on Security and Privacy (SP), pp. 724-738. IEEE, 2019.

Reviewers' Comments:

Reviewer #1:

None

Reviewer #2:

Remarks to the Author:

This revision makes it clearer that the framework proposes a combination of individual existing techniques and federated learning to benefit medical research. The presentation of experimental data has also been made easier to understand.

Reviewer #3:

Remarks to the Author:

I am generally satisfied by the authors' response to the concerns I raised in my initial review.

I appreciate the added algorithmic description in the paper and it provides some additional insights into the technical contribution of this work. I do note that the main techniques described here (e.g., vectorizing operations and taking advantage of SIMD computation in FHE) are fairly standard in the literature on homomorphic encryption. So I still view the main contribution of this paper to be on the systems implementation side.

I am happy to see a more detailed experimental comparison with previous approaches. One thing to note is that the "WAN" setting comparison in Fig. 4 still seems to use a higher-than-usual bandwidth (500 Mbps still seems faster than a typical WAN setting; something around 150 Mbps seems more typical for WAN setting in my opinion). This is a minor point, and I am fine with keeping the comparison as is.

I do have one question regarding the comparison against centralized HE approaches in Table S2 (Supplementary Note 4). How were these estimates computed? I suggest comparing against the paper of Blatt, Gusev, Polyakov, Goldwasser (Secure large-scale genome-wide association studies using homomorphic encryption; PNAS 2020), who propose a GWAS system based on homomorphic encryption. It would also be nice if the paper could discuss/compare the algorithmic designs between this work and the previous work.

Response to Reviewers' Comments for Manuscript NCOMMS-21-09899-T
Entitled: "Truly Privacy-Preserving Federated Analytics for Precision Medicine with
Multiparty Homomorphic Encryption"

We thank the reviewers for their insightful and constructive comments during the whole process. We have carefully reviewed the new comments from the reviewers and addressed them in our revised submission.

REVIEWERS' COMMENTS

Reviewer #2 (Remarks to the Author):

This revision makes it clearer that the framework proposes a combination of individual existing techniques and federated learning to benefit medical research. The presentation of experimental data has also been made easier to understand.

We thank the reviewer for the help in improving our work. We also thank the reviewer for the kind remarks and recognition of the value of our work.

Reviewer #3 (Remarks to the Author):

I am generally satisfied by the authors' response to the concerns I raised in my initial review.

I appreciate the added algorithmic description in the paper and it provides some additional insights into the technical contribution of this work. I do note that the main techniques described here (e.g., vectorizing operations and taking advantage of SIMD computation in FHE) are fairly standard in the literature on homomorphic encryption. So I still view the main contribution of this paper to be on the systems implementation side.

I am happy to see a more detailed experimental comparison with previous approaches. One thing to note is that the "WAN" setting comparison in Fig. 4 still seems to use a higher-than-usual bandwidth (500 Mbps still seems faster than a typical WAN setting; something around 150 Mbps seems more typical for WAN setting in my opinion). This is a minor point, and I am fine with keeping the comparison as is.

We thank the reviewer for her/his initial comments, which were instrumental in order to improve our work. We agree with the reviewer that 500 Mbps can be faster than a typical WAN setting. In our manuscript, we have shown how FAMHE's runtime evolves when halving the bandwidth and doubling the delay, thus enabling the reader to estimate FAMHE's runtime in other specific networks by varying these two parameters.

I do have one question regarding the comparison against centralized HE approaches in Table S2 (Supplementary Note 4). How were these estimates computed? I suggest comparing against the paper of Blatt, Gusev, Polyakov, Goldwasser (Secure large-scale genome-wide association studies using homomorphic encryption; PNAS 2020), who propose a GWAS system based on homomorphic encryption. It would also be nice if the paper could discuss/compare the algorithmic designs between this work and the previous work.

We thank the reviewer for this comment and for pointing out an interesting reference for comparison. We estimated the runtime of a centralized HE-based solution that would follow the same algorithmic approach as FAMHE by including the overheads that would be brought by the use of a centralized bootstrapping that requires more conservative cryptographic parameters. We clarified this in the text of Supplementary Note 5. As suggested by the reviewer, we also compared FAMHE with the work of Blatt et al., who rely on a centralized HE scheme for GWAS. The authors propose two approaches that either exclude the covariates or limit their number and the number of training iterations in order to optimize the achieved runtime. In FAMHE, the covariates are always included and thanks to an efficient use of a multiparty HE scheme, FAMHE is not limited in the number of iterations that it can perform. We expanded Supplementary Note 5 and Supplementary Table 2 to include the suggested comparison.